# Chronotype, Time of Day, and Children’s Cognitive Performance in Remote Neuropsychological Assessment

**DOI:** 10.3390/bs14040310

**Published:** 2024-04-11

**Authors:** Catarina Bettencourt, Luís Pires, Filipa Almeida, Manuela Vilar, Hugo Cruz, José Leitão, Ana Allen Gomes

**Affiliations:** 1Faculty of Psychology and Educational Sciences, University of Coimbra, 3000-115 Coimbra, Portugal; lpires@fpce.uc.pt (L.P.); filipa.almeida.gomes@fpce.uc.pt (F.A.); mvilar@fpce.uc.pt (M.V.); jleitao@fpce.uc.pt (J.L.); a.allen.gomes@fpce.uc.pt (A.A.G.); 2Centre for Research in Neuropsychology and Cognitive Behavioral Intervention, University of Coimbra, 3000-115 Coimbra, Portugal; 3Laboratory of Chronopsychology and Cognitive Systems (ChronCog), University of Coimbra, 3000-115 Coimbra, Portugal; 4Department of Psychology and Education, Faculty of Human and Social Sciences, University of Beira Interior, 6200-209 Covilhã, Portugal; 5Interdisciplinary Research Centre for Education and Development (CeiED), Lusófona University, 1700-284 Lisboa, Portugal; hugo.m.cruz@cm-matosinhos.pt

**Keywords:** chronotype, time of day, cognition, synchrony effect, asynchrony effect, remote neuropsychological assessment, teleneuropsychology

## Abstract

Research on the influence of chronotype and time of day (TOD) on cognitive performance, especially in children, is limited. We explored potential interactive effects, hypothesizing that performance differs when comparing preferred vs. non-preferred TOD. In total, 76 morning-type (MT = 37) or evening-type (ET = 39) children from the third and fourth grades (48.7% girls; M age = 8.05; SD age = 0.51), identified through the Children Chronotype Questionnaire, completed two 30-min neuropsychological assessment sessions via videoconference on the first (9:00) or last hour (16:00) of the school day. The protocol included neuropsychological tests targeting memory, language, and attention/executive domains. The results revealed an interactive effect of medium size between chronotype and TOD on a Rapid Alternating Stimulus (Naming) Task. MT and ET performed faster in asynchrony conditions (morning for ET; afternoon for MT). Additionally, ET outperformed MT in a Backward Digit Span Task, irrespective of TOD. TOD also influenced performance on an Alternating Verbal Fluency Task, with both MT and ET children performing better in the morning. These results underscore the importance of chronotype and TOD in children’s cognitive performance, particularly in working memory and verbal fluency. Children assessed during non-preferred TOD exhibited better performance on some cognitive tasks, challenging the assumption that optimal times always yield superior results.

## 1. Introduction

Most biological functions, as well as several psychological and cognitive processes, present an endogenous circadian rhythm, a pattern of oscillations throughout a period of approximately 24 h. These endogenous rhythms synchronize the internal biological clock with external cues, such as the light–dark cycle [1]. Chronotype refers to the entrainment of the sleep–wake cycle to the phase of the light–dark cycle [1]. Chronotype is influenced by a combination of genetic and environmental factors and plays a critical role in the interindividual differences in rest–activity behavior, performance, and preference for earlier or later hours to engage in cognitive/physical activities [2,3]. Along a continuum of morningness–eveningness, individuals can be categorized as intermediate types, morning types (MTs), or evening types (ETs). Compared to MTs and considering the external clock, ETs tend to prefer later sleep and activity schedules and present later zeniths (i.e., peaks) of physiological circadian markers, such as core body temperature and melatonin secretion [3,4,5,6].

Some authors report that performance in basic and complex cognitive domains is associated with the individual’s chronotype [7]. A meta-analysis on the impact of chronotype on cognition and academic achievement with high-school and university students (total *n* = 2177) reported that eveningness bore a positive relation to the individuals’ cognitive ability, albeit being of small magnitude (*r* = 0.08, *p* = 0.004) [8]. The more prominent circadian differences between ET and MT have been identified in working memory, psychomotor speed, and attention processes, domains connected to frontal lobe activity and that fluctuate differently during the day for distinct chronotypes [2,8,9,10,11]. In tasks that recruit these cognitive domains, ET individuals usually outperform MTs, exhibiting superior working memory capacity [7,10,12,13]. In a study with young adults, the authors report that ETs were more accurate than MTs on a task of visual working memory [7]. Schmidt and colleagues had similar results using a working-memory N-back paradigm, where ETs had a better performance [10]. Specifically with children, a study assessing 5- and 6-year-olds reported similar findings, with ETs having superior performance in two tests that targeted visual working memory [12].

There are daytime-dependent variations in cognitive performance according to the individuals’ chronotype. Although the interplay between chronotype and time of day (TOD) is not fully understood and sometimes presents mixed evidence (e.g., [14]), several studies report synchrony effects, that is, a superior cognitive performance during the preferred/optimal TOD and/or an inferior performance during non-preferred/non-optimal TOD [3,5,15,16,17,18,19]. This implies that, for MTs, their cognitive performance peaks during morning hours, while for ETs, the peak occurs during the afternoon [3]. The literature suggests that this relationship between TOD and chronotype influences performance differently depending on the cognitive processes involved in the considered task. Tasks targeting attention [20], working memory [15,21], fluid intelligence [22], and verbal fluency [23] appear to exhibit synchrony effects. Synchrony effects have mainly been seen in tasks with higher difficulty and complexity, involving controlled efforts to process and retrieve information [3,10,21,24,25]. Paradoxically, the opposite phenomenon has also been reported, as not all research has found exclusively synchrony effects. Some tasks are performed better at non-optimal/non-preferred TOD, reflecting an asynchrony effect [7,26,27,28]. That means that, for MTs, their cognitive performance peaks during the afternoon, while for ETs, the peak occurs during the morning. Asynchrony effects are reported in tasks involving spatial intelligence [29], implicit memory and learning [21,25], and creative decision-making processes [27,30]. More recently, a study with adults *(n* = 324) exploring eyewitness memory performance found that participants were 1.6 times more likely to give an accurate answer at non-preferred compared to preferred TOD [31]. These findings imply that performance on tasks requiring automatic processes, such as face perception and recognition, seems better during non-preferred TOD. Automatic processes may benefit from non-preferred TOD possibly due to a decrease in the efficiency of cognitive control processes during off-peak hours, specifically inhibition [5,12,13,21,25,32].

There are also observable TOD effects, that is, tasks that are performed differently in morning or afternoon hours. Research involving both children and adults has indicated that working memory tends to be more efficient in the morning, whereas long-term declarative memory shows enhanced performance later in the afternoon [11,12,13,33,34,35,36]. A study with young adults found significant differences in declarative memory performance in a list-recalling task, with participants generally recalling more words in the evening [2]. Wilks et al. [37] assessed associative memory, processing speed, and visuospatial working memory in older adults (*n* = 169; aged 61–94 years). The results revealed TOD effects where participants performed approximately 10% better in morning hours compared to evening hours.

Although scarce, the majority of studies on the interplay chronotype x TOD in cognitive function is conducted with adult samples. Data on school-aged children and adolescents are even scarcer (e.g., [38]), thus limiting our knowledge on how these factors affect children’s cognitive assessments. Most studies on cognition with pediatric populations focus on sleep patterns or TOD or chronotype effects independently, not in interaction [38,39,40,41,42]. On top of the fact that the combined influence of TOD and individual differences on chronotype is still unclear in school-aged children, assessments via Teleneuropsychology (TeleNP) in children remain mostly unexplored. In TeleNP, professionals and clients/participants interact through telecommunication technologies, such as secure videoconference software [43]. Such remote assessments have gained particular importance due to COVID-19 pandemic regulations encouraging physical distancing and are becoming perceived as a valuable resource whenever face-to-face encounters are not feasible. Accumulated empirical evidence on TeleNP has supported the feasibility, reliability, validity, and acceptability of administering neuropsychological tests remotely in a variety of contexts for both adult and pediatric populations [43,44,45,46,47,48,49,50,51]. TeleNP and in-person assessment scores have shown no significant differences [43,46,47,48,49,51,52,53,54]. A systematic review conducted by Ruffini et al. [43] on the comparison of TeleNP and in-person assessment (23 studies, N = 2193 children, targeting language, memory, and executive functions) reported no significant differences between the two methods of administration, taking into account the type of tasks and stimuli used [43]. Verbally administered measures and tasks relying primarily on verbal responses are particularly suited for remote administration [47,48]. Some support was also found for remote administration of tasks relying on visual stimuli that can be presented on the screen during TeleNP [53,55,56].

Most of the research in TeleNP assessment with children has been conducted in the context of primary healthcare or home-based settings. In this study, we assessed children within the school environment, during their regular classroom activities. Most schools provide instruction via remote learning, and children and school staff are increasingly familiar with videoconferencing technology [57]. To the best of our knowledge, there is very little research conducted in ecological school settings. Given the paucity of studies with TeleNP on children, as well as on the impact of chronotype and TOD on children, we aimed to investigate the potential interactive effects of chronotype and TOD on children’s cognitive performance (i.e., memory, language, and executive functions) with TeleNP assessments. We hypothesized that cognitive performance differs when comparing preferred vs. non-preferred TOD depending on the profile of cognitive processes that are recruited by the task at hand (i.e., the task-set). Specifically, we anticipated synchrony effects when the quality of task performance univocally reflected the efficiency of controlled processes, and asynchrony effects when the task-set related to controlled and automatic processes, the latter either being suppressed or having its outputs inhibited by a controlled process, while the quality of task performance does in fact benefit from those automatic processes.

## 2. Materials and Methods

### 2.1. Participants

Third and fourth graders were recruited from 15 primary schools in Central Portugal. From an initial pool of 483 children, 76 Portuguese-native speakers between the ages of 7 and 10 years old were included in the study (37 females; *M* age = 8.05; *SD* age = 0.51). ET and MT children were selected based on their chronotype, which was assessed with the Children’s Chronotype Questionnaire (CCTQ) [58,59]. The remaining children were not invited to participate, including those with special needs or other conditions potentially affecting performance (*n* = 8), as well as those classified as intermediate chronotypes (*n* = 399). This decision was made to ensure focus on the effects of chronotype and TOD on cognitive performance, deliberately selecting participants at the extreme ends of the chronotype continuum to better discern potential subtle effects. Ultimately, we included in the study 37 children classified as MTs (17 females; *M* age = 8.13; *SD* age = 0.08) and 39 as ETs (20 females; *M* age = 7.97; *SD* age = 0.08) with CCTQ. Among the 39 ET children, based on randomized allocation, 20 underwent assessment in the morning (9:00) and 19 in the afternoon (16:00). Out of the 37 MT children, based on randomized allocation, 19 were assessed in the morning (9:00) and 18 in the afternoon (16:00). MTs and ETs did not differ in age [*t*(74) = 1.32, *p* = 0.19] or sex [*χ*^2^(1, 76) = 0.22, *p* = 0.64], but they differed in sleep period during school days [*t*(71) = −2.54, *p* = 0.013], as expected. Participants evaluated during morning and afternoon hours did not differ in age [*t*(74) = 0.42, *p* = 0.68], sex [*χ*^2^(1, 76) = 0.85, *p* = 0.36], or sleep period during school days [*t*(71) = −0.04, *p* = 0.97]. The groups of MTs and ETs tested either in the morning or the afternoon were similar in age, sleep period during school days, and sex, except for ETs who revealed an imbalance in sex between the two conditions (morning vs. afternoon).

### 2.2. Measures

All TeleNP assessment instruments were adapted versions of standardized paper-and-pencil tests (as specified below), optimized for remote administration. Only tasks requiring verbal responses were employed, excluding all tasks for which motor responses were necessary. Participants responded through a microphone during the videoconference, using a shared-screen feature. Standard neuropsychological assessment procedures were only partially modified to allow for remote administration with school-aged children.

#### 2.2.1. Children’s Chronotype Questionnaire (CCTQ)

Chronotype was assessed through a parental-report paper and pencil test, the CCTQ [58,59]. CCTQ is a mixed-format questionnaire for 4- to 11-year-olds. Children’s chronotype was determined using the morningness–eveningness (M/E) scale, where higher scores represent higher eveningness. Additionally, their score was compared and considered, along with their response to question 27, a single inquiry about ‘chronotype’, ensuring congruence between the two (cf, [58,59]). Children displaying scores on the M/E scale falling below the 20th percentile or above the 80th percentile (if not incongruent with item 27’s answer) were categorized as morning and evening types, respectively. These cutoff points were a conservative adaptation of the Horne and Östberg [60] criteria and have been employed in previous studies [61,62]. Reliability analysis for the morningness–eveningness scale yielded a good Cronbach’s Alpha of *α* = 0.87 [63].

#### 2.2.2. Verbal Fluency

Verbal Fluency assessment included four types of tasks: Free-Word Fluency Task, Semantic (or Category) Fluency Task, Phonemic (or Letter) Fluency Task, and Alternating Fluency Task. In the Free-Word Fluency Task, participants were asked to generate as many words as possible in 60 s, without fixed accuracy criterion [64]. In the Semantic Fluency Task, participants generated as many words as possible in 60 s based on a semantic category (two trials: animals and food) [65]. During the Phonemic Fluency Task, participants were tasked with generating as many words as possible in 60 s, starting with a specified letter. Proper names and morphological derivations of the same word were not allowed (two trials: P and M) [65]. The Alternating Fluency Task required participants to set-shift within two different semantic categories (two trials: fruits/furniture and clothing/colors) [66]. Participants answers were recorded to ensure accurate scoring. Each acceptable word was given 1 point. The number of correct trials of different types within each task was added to obtain overall scores for Semantic Fluency, Phonemic Fluency, and Alternating Fluency. For the Alternating Fluency Task, the number of successful switches between categories was also calculated. Regarding reliability analyses, for the Semantic Fluency and Phonemic Fluency Tasks, the split-half approach yielded adequate scores of *r*_sb_ = 0.73 and *r*_sb_ = 0.69 for internal consistency, respectively. For the Alternating Fluency Task, a lower score of *r*_sb_ = 0.42 was obtained.

#### 2.2.3. Digit Span

The Digit Span subtest from the Wechsler Intelligence Scale for Children—Third edition (WISC-III) [67,68] includes Digit Span Forward and Digit Span Backward. Digit Span Forward requires participants to repeat verbally presented digit sequences, while Digit Span Backward requires participants to reproduce the sequences in reverse order (i.e., from the last to the first digit presented). Digit Span Forward has eight increasing levels of difficulty, from two-digit up to nine-digit sequences. Digit Span Backward has seven increasing levels of difficulty, from two-digit up to eight-digit sequences. Each level comprises two trials of the same length. Participants must correctly reproduce at least one trial at each level to proceed to the next one. The task is discontinued when both trials are failed. Each correct sequence is given 1 point (maximum 16 points for Digit Span Forward and 14 for Digit Span Backward). The totals of correct trials at each difficulty level are summed to calculate the Digit Span Forward and the Digit Span Backward scores. Regarding reliability analyses in our study, for the Digit Span Forward and Backward Tasks, the split-half approach yielded adequate scores of *r*_sb_ = 0.74 and *r*_sb_ = 0.72 for internal consistency, respectively.

#### 2.2.4. Face Recognition Task

The Face Recognition Task from the Coimbra Neuropsychological Assessment Battery (BANC) [69] includes three parts. In the learning phase, participants are instructed to remember 16 black-and-white photographs of faces with neutral expressions (eight males and eight females). The stimuli are presented on a white background with a 3000-millisecond delay between them. Subsequently, in the Immediate Recognition trial, 16 sets of three photographs are presented for the participants to identify which one they saw previously. After 20 to 30 min, in the Delayed Recognition Trial, the same 16 sets of three photographs were again presented in a different order for participants to identify which one they had seen in the learning phase. Each correctly selected photograph is given 1 point (maximum 16 points). The respective totals of correctly identified photographs in the Immediate and Delayed Recognition Trials are calculated. Regarding reliability analyses, the split-half approach yielded a good score for internal consistency both for the Immediate (*r*_sb_ = 0.79) and for the Delayed Recognition Trials (*r*_sb_ = 0.81).

#### 2.2.5. Story Memory Task

The Story Memory Task from BANC [69] comprises two story scripts that are read aloud to the participant. Participants are asked to freely recall both stories immediately after presentation (Immediate Recall Trial) and after a 20-to-30-min delay (Delayed Recall Trial). Scoring is completed through a standardized scoring template with 36 pre-defined information units representing different script elements. After the Delayed Recall Trial of both stories, a Recognition Trial comprising 30 multiple-choice questions is completed for both scripts. Each correctly recalled piece of information unit is given 1 point (maximum 36 points), as well as each correct option in the Recognition Trial (maximum 30 points). The scores from both the Immediate and Delayed Recall Trials are summed to obtain overall Immediate and delayed recall scores, and the correct answers from both recognition trials are summed for an overall recognition score. The retention score is presented as a percentage determined by dividing the overall delayed recall score by the overall immediate recall score and multiplying by 100. Reliability analyses using the split-half method estimated good scores of *r*_sb_ = 0.69 and *r*_sb_ = 0.71 for immediate and delayed recall, respectively. For recognition, an acceptable score of *r*_sb_ = 0.67 was obtained.

#### 2.2.6. Rapid Automatized Naming Task and Rapid Alternating Stimulus Task

In the Rapid Automatized Naming and Rapid Alternating Stimulus Tasks from BANC [69], participants are asked to accurately name, as quickly as possible, an array of familiar visual stimuli forming a 10 × 5 matrix in a white background, in left-to-right serial fashion. In the Rapid Automatized Naming Task, participants were required to name numbers (5 possible numbers: 2, 4, 6, 7, and 9). The Rapid Alternating Stimuli Task contained two types of alternating stimuli (geometric shapes and colors) that must be named, e.g., red triangle (4 possible geometric shapes, triangle, square, circle, and rectangle; 4 possible colors, red, red, yellow, and green). These two tasks are similar, but the Rapid Alternating Stimulus Task is more challenging, as it requires a continuous set-shift between the stimulus characteristics of the item that must be named. The total time (in seconds) spent naming all items in the Rapid Automatized Naming and Rapid Alternating Stimulus Tasks is recorded. The Test–Retest Reliability from the original studies was good for the Rapid Alternating Stimulus Task (*r*(67) = 0.90, *p* < 0.01) and Rapid Automatized Naming Task (*r*(67) = 0.78, *p* < 0.01). Due to these tasks’ nature, no estimates of internal consistency can be obtained for these tasks.

#### 2.2.7. Language Comprehension Task

The Language Comprehension Task from the BANC [69] is composed of picture selection trials pertaining to 27 sentences read aloud. The participant listens to a sentence and selects the numbered picture out of 9 that corresponds to the sentence. The 27 sentences are organized into 3 sets of 9 items each. The difficulty increases within each set. Each correctly selected picture is given 1 point. The test score was computed as the total of correct responses (maximum 27). Regarding reliability analyses in our study, the split-half approach yielded a good score of *r*_sb_ = 0.81.

### 2.3. Procedures

Data collection took place from April to July 2021 (spring and summer months in the Northern Hemisphere), following the second COVID-19 mandatory confinement period in Portugal. All participants were enrolled in 3rd or 4th grade of 15 elementary schools in Central Portugal. COVID-19 physical-distancing measures restricted access to the school perimeter for non-school staff members. Teachers served as intermediaries for all necessary communication with parents/caregivers, and all participant contact occurred through videoconferencing.

After written informed consent, parents/caregivers filled a paper-and-pencil version of CCTQ, used to determine the participants’ chronotype based on scores on the M/E scale and on item 27 (as detailed previously). Afterward, MT and ET children were selected and invited to complete the TeleNP assessment. To mitigate the potential impact of order effects, the neuropsychological protocol order was counterbalanced, totaling 16 different possible orders. The protocol was administered in two 30-min online sessions, separated by a 7-day interval. All participants completed the sessions with a laptop or desktop. TeleNP assessments were conducted during class schedule either on the first or last hour of the school day (09:00 h or 16:00 h), based on randomized allocation. About half of the MTs were assessed in the morning (*n* = 19) and the other half in the afternoon (*n* = 18). About half of the ETs were assessed in the morning (*n* = 20) and the other half in the afternoon (*n* = 19).

Sessions were conducted in classrooms, with natural daylight exposure and minimal distractions. Only the participant and a teacher/educational assistant were physically present. Teachers/assistants were present to help with the setup and any technical difficulty and did not otherwise interfere with the course of the assessment. Participants were briefed about potential technological issues, such as Wi-Fi disconnection or audio difficulties, and were instructed to re-initiate the Zoom session in the case of any technical problem. Teachers/assistants were given the examiner’s phone number to contact if technical issues persisted. Wi-Fi connection and video and audio quality were tested beforehand, and sessions were postponed if quality criteria were not met. TeleNP sessions were conducted using the same videoconference software (Version number 5.16.10, 26186) to reduce potential bias. We used Zoom—Professional Version, a secure and approved cloud-based platform. Zoom is amongst the most widely used platforms for teleconferencing, and in some health systems, it has been approved for clinical interactions, as stated by international multiorganization guidelines [70,71]. With the authors’ permission, selected visual stimuli were presented using Zoom’s screen-share feature, enabling participants to view stimuli shared from the examiner’s computer. Examiners delivered the instructions verbally and presented test items visually or verbally to the participant. Children were instructed to position themselves in a manner that allowed for a full view of their faces and hands. This ensured that examiners could confirm that they were not engaging in behaviors that compromised the test standardization and validity, such as writing down test items.

### 2.4. Statistical Analysis

All analyses were performed using version 25.0 of IBM SPSS Statistics for Windows (IBM Corp., Armonk, NY, USA). A significance level of α = 0.05 was considered for all statistical tests. Reliability for each neuropsychological test was estimated using the split-half approach and corrected with the Spearman–Brown prediction formula [72,73]. For the main statistical analysis, 2 × 2 ANOVAs were performed, one for each dependent variable (indexes and totals of all cognitive tests used), with chronotype (MT and ET) and TOD (morning/afternoon) as independent variables. In the case of a significant interaction, post hoc comparisons with a Bonferroni adjustment for multiple comparisons were performed within each level of chronotype (MT and ET) to examine the differences between morning and afternoon performance. To evaluate magnitude, the classification of partial eta squared ($\eta_{p}^{2}$) for each effect followed Cohen’s (1988, 284–287) criteria for Eta square, where η^2^ = 0.01 is a small effect size, η^2^ = 0.06 is a medium effect size, and η^2^ = 0.14 is a large effect size. Descriptive statistics were computed to characterize the sample on all tests’ total scores: means (*M*) and standard deviations (*SD*) are provided for each test score (see Table 1 and Table 2).

## 3. Results

Descriptive statistics for all relevant test scores, as well as age, morningness–eveningness, and sleep period, in MT and ET children, are shown on Table 1 and Table 2. Sleep period on school (scheduled) days was estimated as the difference between sleep onset time and morning wakeup time, using the Children’s Chronotype Questionnaire (CCTQ) [58,59]. Analyses of variance (2 × 2 ANOVAs) yielded significant chronotype and TOD main or interactive effects in specific indexes of three sub-tests: Rapid Alternating Stimulus Task, Backward Digit Span Task, and Alternating Verbal Fluency Task.

The results showed a significant interactive effect of medium size between chronotype and TOD on the time to complete the Rapid Alternating Stimulus Task [*F* (1, 72) = 5.78, *p* = 0.019, $\eta_{p}^{2}$ = 0.07]. MTs took significantly less time (in seconds) to complete the task in the afternoon (*M* = 110.72; *SD* = 27.15) than in the morning (*M* = 124.58; *SD* = 50.03). ETs took significantly less time (in seconds) to complete the task in the morning (*M* = 111.55; *SD* = 26.05) than in the afternoon (*M* = 146.00; *SD* = 61.21). This pattern constitutes an asynchrony effect: When tested at their non-preferred TOD (i.e., morning for ETs, and afternoon for MTs), MT and ET children were faster to complete the task than at their preferred TOD (see Figure 1).

There was also a chronotype main effect of medium size on the total score of the Backward Digit Span Task [*F* (1, 72) = 5.98, *p* = 0.017, $\eta_{p}^{2}$ = 0.08]. ET children (*M* = 5.23; *SD* = 1.74) scored higher than MT children (*M* = 4.32; *SD* = 1.45), successfully completing more trials (see Figure 2).

Additionally, there were medium-to-large main effects of TOD on two indexes of the Alternating Verbal Fluency Task, namely the total score [*F* (1, 72) = 6.23, *p* = 0.015, $\eta_{p}^{2}$ = 0.08] and total of successful switches [*F* (1, 72) = 8.85, *p* = 0.004, $\eta_{p}^{2}$ = 0.11]. All participants, irrespective of chronotype, scored higher in the morning (*M* = 21.44; *SD* = 4.85) than in the afternoon session (*M* = 18.86; *SD* = 3.95). Likewise, both MTs and ETs completed more successful switches between two semantic categories in the morning (*M* = 25.23; *SD* = 5.39) than in the afternoon session (*M* = 21.62; *SD* = 5.04) (see Figure 3 and Figure 4).

## 4. Discussion

The current investigation is one of the first studies to evaluate the effects of chronotype and TOD on children’s cognition using remote school-based administration of a neuropsychological battery. Significant main or interactive effects were detected in specific indexes of three sub-tests: Backward Digit Span Task, Alternating Verbal Fluency Task, and Rapid Alternating Stimulus Task. ETs outperformed MTs in a Backward Digit Span Task, regardless of TOD. TOD influenced performance on an Alternating Verbal Fluency Task, with both MT and ET children performing better in the morning. An interaction effect (specifically an asynchrony effect) was observed on a Rapid Alternating Stimulus Task, where both MTs and ETs exhibited faster performance during non-preferred TOD (morning for ETs, and afternoon for MTs).

Concerning the influence of chronotype on cognition, ETs outperformed MTs on a task involving auditory working memory, where children are asked to recall a sequence of numbers in reverse order. In line with our results, other studies have found superior working memory performance for ETs, scoring higher than MTs even when these cognitive tasks are performed in the morning [7,10,12,13]. Specifically with children, a study assessing 5- and 6-year-olds found that ETs had superior performance in tests that targeted visual working memory [12]. Despite the absence of a consensual explanation for the superiority of ETs, Preckel et al. [5] propose that the challenges ETs face in adjusting to the school schedule, typically starting early in the morning, which conflicts with their internal biological clock, might contribute to the development of enhanced problem-solving abilities. There is a relationship between working memory capacity and problem-solving abilities. Working memory plays a role in establishing and maintaining attentional focus and in resisting distractions, which helps the process of narrowing down the solution space during problem-solving [74]. One might argue that, as individuals practice and develop their problem-solving abilities, there is also an improvement in working memory. This could explain why we found that ETs performed better than MTs in a task involving auditory working memory.

Our findings also revealed a significant main effect of TOD, where performance on a verbal fluency task was superior in the morning, worsening in the afternoon. This particular task (Alternating Verbal Fluency Task) is more demanding on working memory than other existing variations. Children are required to produce words from two different semantic categories alternately, while keeping track of the words they have already uttered. Similarly to other psychological and cognitive functions, memory manifests circadian oscillations, with acrophases at different times over the 24-h cycle, depending on the type of memory considered. Studies with both children and adults show that short-term memory performance peaks in the morning, while long-term memory is better in the afternoon [11,33,34,35]. Moreover, alternating verbal fluency does not only depend on language production abilities, as regular verbal fluency tasks, but also actively recruits working memory, namely for tracking prior utterances. This accrued recruitment of working memory and the reported superiority of short-term memory in the morning likely account for the superior alternating verbal fluency in the morning relative to the afternoon.

A significant chronotype and TOD interactive effect, specifically an asynchrony effect, was also uncovered in a naming task, the Rapid Alternating Stimulus Task. Both MTs and ETs performed faster at their non-preferred TOD (morning for ETs; afternoon for MTs). As previously discussed, this interactive effect could be examined in light of the profile of cognitive processes that are being recruited. Performance on naming tasks requires a level of automaticity and multiple interconnected processes, as stated by the theory of multiple constructs by Wolf and Bowers [75]. These tasks primarily involve lexical access processes, where the lexical–semantic representation is linked to its orthographic representation for written naming [75,76,77]. The performance also relies on the efficiency of other interconnected processes, such as global processing speed and attentional, visual, and articulatory processes [69,75,76,78,79]. This cognitively complex task involves controlled as well as automatic processes, such as the automatic activation of task-appropriate lexical information [75,80,81]. In the particular case of the naming task we used, participants must continuously alternate between naming two semantic categories/stimuli characteristics: geometric shapes and colors. Participants may present a task-set that strictly translates the given instruction of alternating between naming the geometric shape and naming the color. Alternatively, they may approach the task with the sole goal of producing adjectival units (e.g., ‘red triangle’, where a noun is modified by an adjective to describe the shape as being red, instead of separately naming the color and then the shape—‘red’ and ‘triangle’), which speeds up the production of the answer. Consistent with the theory outlined earlier, which considers the cognitive processes involved [12,21,25,32], one could argue that, during the preferred TOD, it is more likely for participants to adhere closely to the given instructions. This requires controlled efforts by the executive function to switch between naming the shape and naming the color. Conversely, during non-preferred TOD, the application of the syntactic rule that generates adjectival units—an automatic process—may not be inhibited by the executive process responsible for alternating between the objectives of naming the shape and naming the color. Therefore, accurate responses driven by these automatic processes are more likely to be produced at the non-preferred TOD, without being hampered by cognitive control processes [12,21,32]. The quality of task performance benefits from those automatic processes, thereby leading to increased speed and, consequently, superior performance during non-preferred TOD.

While we were able to identify the main and interactive effects, namely asynchrony, a majority of the tasks employed in our study appear to be unaffected by chronotype or TOD. The literature suggests that the absence of chronotype and TOD effects is particularly notable in tasks that involve crystallized intelligence (e.g., vocabulary tasks) or tasks where the predominant response for the majority of individuals is correct [16,29,82,83]. In fact, we found no effects on the Language Comprehension Task, a test that recruits crystalized intelligence, as participants draw upon previously acquired knowledge to solve problems. This observation can also account for the lack of effects on the Rapid Automatized Naming Task, a relatively straightforward task where participants are required to name numbers.

This study differs from prior TeleNP studies, as the remote assessment took place at the children’s school rather than in a clinical or domiciliary setting, providing a more ecologically valid context. These findings highlight the significance of considering chronotype and TOD in the context of children’s cognitive performance, specifically within the cognitive domains of working memory and verbal fluency. Depending on the specific cognitive function recruited, children assessed at non-preferred TOD occasionally displayed superior performance, challenging the assumption that performing evaluations during preferred TOD always results in better outcomes.

Nevertheless, our study presents a few limitations. A larger sample size would enhance the study results’ robustness. However, the constraints imposed by the COVID-19 pandemic and the complexity of data collection limited the feasibility of expanding the sample size. The post hoc analysis revealed a power of 0.732. Our study was carried out after a period of mandatory confinement, during which physical-distancing measures restricted access to the school perimeter solely to school staff members. This circumstance rendered in-person neuropsychological assessments logistically unfeasible. An additional potential limitation of our study was the employment of a between-subject design (i.e., children were tested either at their optimal or non-optimal TOD). Although we used randomized allocation, one might argue that a within-subject design might have better controlled for potentially relevant inter-individual differences. Lastly, the assumption of similar engagement in TeleNP assessments across all participants may overlook individual variations in technology familiarity, which could potentially influence the study’s outcomes. However, in the Portuguese school system, children are introduced to and actively engage with computers in the second grade. Despite their familiarity with technology, it was the teachers who were responsible for setting up the computers, while children only had to observe stimuli on the screen and provide verbal responses to the tasks. Despite these limitations, this controlled study provides valuable insights into the interplay of chronotype and TOD on children’s cognitive performance in a real-world, school-based context.

Further research can advance the comprehension of children’s chronopsychology, offering insights for both education and research. Future investigations should consider a broader age range, encompassing various developmental stages, to capture age-related differences in the interaction between chronotype, TOD, and cognitive performance. Expanding the age groups to include both pre- and post-pubertal participants would help to better understand the processing mechanisms underlying synchrony versus asynchrony effects. It is well-documented that controlled processes improve throughout childhood [84,85], and the educational environment plays a role in enhancing the level of task automatization for tasks recruiting scholastic skills. Moreover, children’s circadian rhythm is typically set to earlier times compared to adolescents, with a phase delay occurring during puberty [86,87,88]. Consequently, it cannot be safely assumed that synchrony or asynchrony effects observed at a specific developmental stage will be present at later stages. Lastly, the use of objective measures of chronotype, such as actigraphy or physiological markers, in addition to chronotype questionnaires, could enhance the precision of assessments in future studies.

## Figures and Tables

**Figure 1 behavsci-14-00310-f001:**
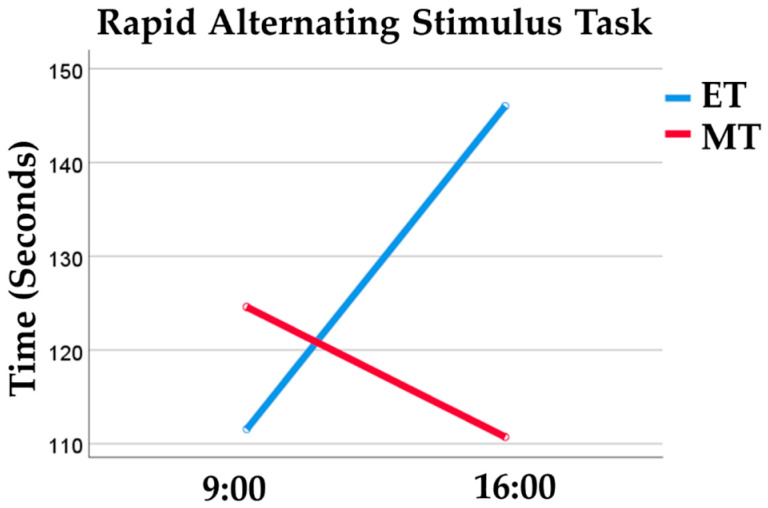
Performance of morning (MT) and evening (ET) chronotypes during the Rapid Alternating Stimulus Task. The top graph illustrates the mean time (in seconds) taken by each group to complete the task in both morning and afternoon sessions. The bottom graph represents the 95% Confidence Interval error bars.

**Figure 2 behavsci-14-00310-f002:**
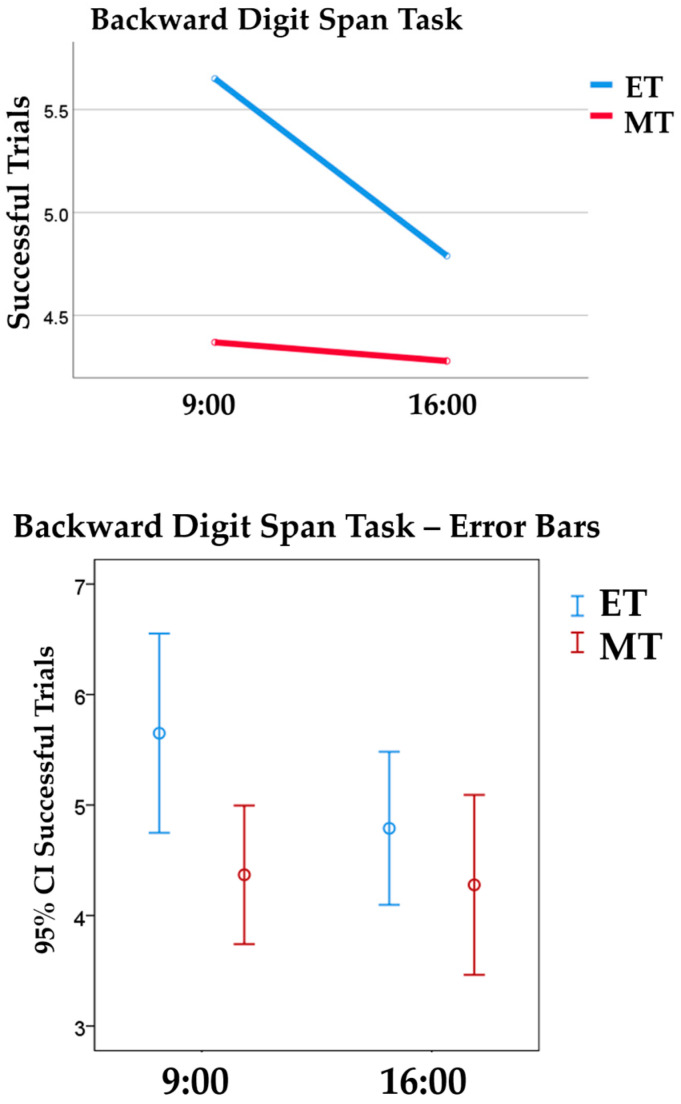
The top graph represents morning (MT) and evening (ET) chronotypes’ performance on the Backward Digit Span Task, measured by successful trials, in both morning and afternoon sessions. The bottom graph represents the 95% Confidence Interval error bars.

**Figure 3 behavsci-14-00310-f003:**
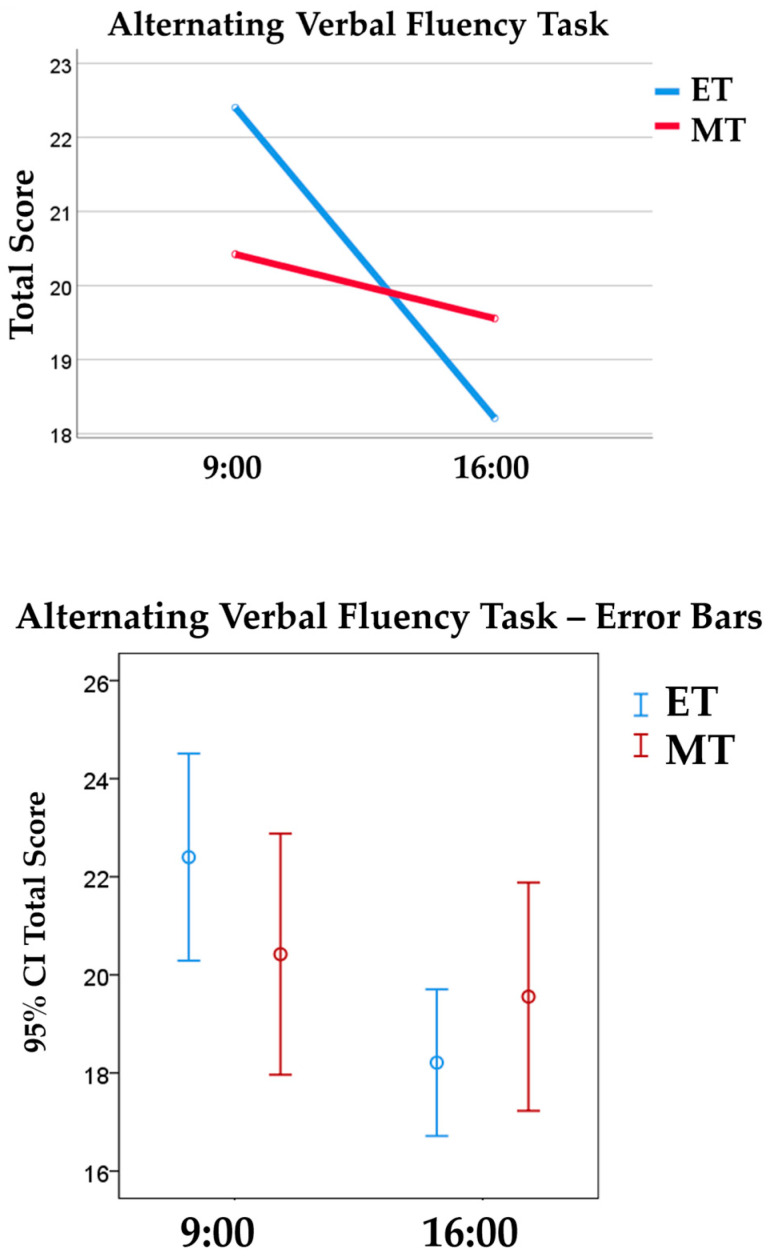
The top graph illustrates morning (MT) and evening (ET) chronotypes’ performance on the Alternating Verbal Fluency Task in both morning and afternoon sessions, measured by total score (total of correct words). The bottom graph represents the 95% Confidence Interval error bars.

**Figure 4 behavsci-14-00310-f004:**
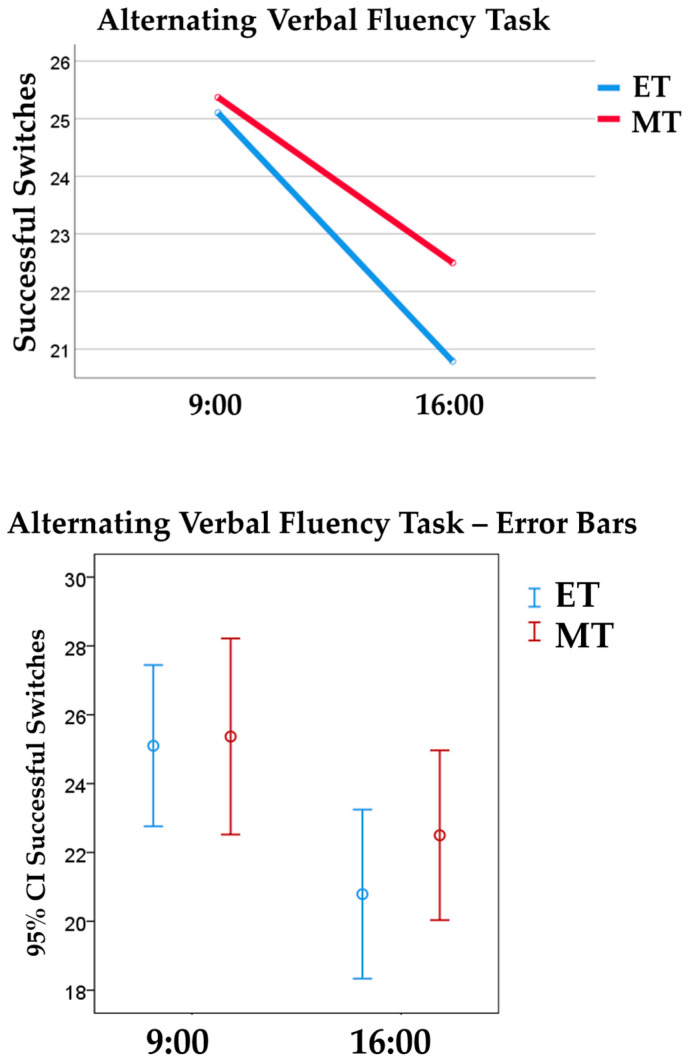
The top graph represents morning (MT) and evening (ET) chronotypes’ performance on the Alternating Verbal Fluency Task in both morning and afternoon sessions, measured by successful switches between two semantic categories (e.g., fruits and furniture) in 60 s. The bottom graph represents the 95% Confidence Interval error bars.

**Table 1 behavsci-14-00310-t001:** Age, morningness–eveningness score, sleep period, and relevant neuropsychological test scores by chronotype (MT or ET children).

Variable	M-Types (*n* = 37) *M* (*SD*)	E-Types (*n* = 39) *M* (*SD*)	Total (*n* = 76) *M* (*SD*)	Significance *p*
Age (years)	7.97 (0.50)	8.13 (0.52)	8.05 (0.51)	n.s.
ME scale	25.18 (0.34)	39.54 (0.51)	-	*p* < 0.001
Sleep period—school days (hours)	9.69(0.57)	9.32(0.65)	9.50(0.64)	*p* = 0.013
Semantic Verbal Fluency (total)	31.24 (9.56)	31.67 (9.68)	31.46 (9.56)	n.s.
Phonemic Verbal Fluency (total)	12.68 (5.01)	10.69 (3.88)	11.66 (4.55)	n.s.
Free-Word Fluency (total)	28.51 (8.43)	28.28 (10.91)	28.39 (9.72)	n.s.
Alternating Verbal Fluency (total)	20.00 (4.85)	20.36 (4.39)	20.18 (4.59)	n.s.
Alternating Verbal Fluency (successful switches)	23.97 (5.59)	23.00 (5.44)	23.47 (5.50)	n.s.
Digit Span Forward (total)	7.27 (1.73)	7.41 (1.68)	7.34 (1.69)	n.s.
Digit Span Backward (total)	4.42 (1.45)	5.23 (1.74)	4.79 (1.66)	*p* = 0.017
Face Recognition—Immediate Recall (total)	12.32 (2.47)	12.05 (2.42)	12.18 (2.43)	n.s.
Face Recognition—Delayed Recall (total)	12.65 (2.95)	12.28 (2.69)	12.46 (2.80)	n.s.
Story Memory—Immediate Recall (total)	38.46 (14.00)	40.31 (10.51)	39.41 (12.29)	n.s.
Story Memory—Delayed Recall (total)	36.81 (13.86)	38.56 (11.37)	37.71 (12.59)	n.s.
Story Memory—Recognition (total)	25.35 (3.51)	26.18 (3.14)	25.78 (3.33)	n.s.
Story Memory—Retention (percentage)	95.35 (10.93)	95.18 (13.59)	95.26 (12.28)	n.s.
Rapid Automatized Naming Task—Time (seconds)	30.84 (10.16)	29.92 (4.91)	30.37 (7.87)	n.s.
Rapid Alternating Stimuli Task—Time (seconds)	117.84 (40.60)	128.33 (49.18)	123.22 (45.22)	n.s.
Language Comprehension (total)	18.03 (4.86)	18.26 (4.21)	18.14 (4.51)	n.s.

A significance level of α = 0.05 was considered for all statistical tests. n.s. = non-significant.

**Table 2 behavsci-14-00310-t002:** Relevant neuropsychological test scores by TOD (morning or afternoon).

Variable	Morning (*n* = 39) *M* (*SD*)	Afternoon (*n* = 37) *M* (*SD*)	Significance *p*
Semantic Verbal Fluency (total)	31.64 (10.23)	31.27 (8.94)	n.s.
Phonemic Verbal Fluency (total)	12.05 (4.92)	11.24 (4.15)	n.s.
Free-Word Fluency (total)	27.64 (7.79)	29.19 (11.47)	n.s.
Alternating Verbal Fluency (total)	21.44 (4.85)	18.86 (3.95)	*p* = 0.015
Alternating Verbal Fluency (successful switches)	25.23 (5.39)	21.62 (5.04)	*p* = 0.004
Digit Span Forward (total)	7.03 (1.58)	7.68 (1.77)	n.s.
Digit Span Backward (total)	5.03 (1.76)	4.54 (1.54)	n.s.
Face Recognition—Immediate Recall (total)	12.38 (2.31)	11.97 (2.57)	n.s.
Face Recognition—Delayed Recall (total)	12.92 (2.70)	11.97 (2.86)	n.s.
Story Memory—Immediate Recall (total)	40.21 (13.25)	38.57 (11.18)	n.s.
Story Memory—Delayed Recall (total)	38.10 (13.22)	37.30 (12.06)	n.s.
Story Memory—Recognition (total)	25.41 (3.62)	26.16 (2.99)	n.s.
Story Memory– Retention (percentage)	94.44 (12.47)	96.13 (12.19)	n.s.
Rapid Automatized Naming Task—Time (seconds)	30.44 (9.04)	30.30 (6.55)	n.s.
Rapid Alternating Stimulus Task—Time (seconds)	117.90 (39.60)	128.84 (50.41)	n.s.
Language Comprehension (total)	18.31 (4.73)	17.97 (4.32)	n.s.

A significance level of α = 0.05 was considered for all statistical tests. n.s. = non-significant.

## Data Availability

Data are available upon reasonable request to the corresponding author, C.B.

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
