# Peer review of "Chronotype, Time of Day, and Children’s Cognitive Performance in Remote Neuropsychological Assessment"

_behavsci, 2024, doi:10.3390/bs14040310_

Round 1
Reviewer 1 Report
Comments and Suggestions for Authors
“Chronotype, time-of-day, and children's cognitive performance in remote neuropsychological assessment” examined the interactive effects of chronotype and time-of-day on cognitive performance in children. The results reveal interactive effects of chronotype and time-of-day on a naming task. Interestingly, there was also a performance advantage for evening-type over morning-type on the digit span task regardless of the time-of-day and both types of children performed better on the verbal fluency task in the morning compared to the evening. Overall, I found the manuscript nicely crafted and likely beneficial to the audience of Behavioral Science; however, prior to publication, a minor inclusion of development is needed in the discussion section. The authors briefly mention broadening age as a future direction. A brief discussion on how development may have contributed to findings will assist the reader in understanding the limits and practical implications of the findings.
Author Response
The authors wish to thank the opportunity to strengthen the manuscript with valuable comments and queries from the Reviewers. We value the careful attention and constructive feedback on our original submission titled “Chronotype, time-of-day, and children's cognitive performance in remote neuropsychological assessment”. Amendments were incorporated to reflect the detailed suggestions the Reviewers have provided. We hope that our edits and the responses provided below adequately address all the issues and concerns the Reviewers have noted.
Reviewer #1
Comment: Overall, I found the manuscript nicely crafted and likely beneficial to the audience of Behavioral Science; however, prior to publication, a minor inclusion of development is needed in the discussion section. The authors briefly mention broadening age as a future direction. A brief discussion on how development may have contributed to findings will assist the reader in understanding the limits and practical implications of the findings.
Response: We appreciate the positive feedback on the overall quality of the manuscript, as well as the insightful suggestion regarding the discussion section. In the revised manuscript, we have included a brief explanation on how the development of cognitive abilities from childhood to adolescence, particularly in relation to executive function, and how the puberty-induced changes in circadian preferences may impact the manifestation of synchrony or asynchrony effects. We believe that this addition has enhanced the discussion, providing greater clarity regarding the limitations and practical implications of the results. We extend our gratitude to Reviewer #1 for the valuable feedback and assistance in enhancing our manuscript.
Reviewer 2 Report
Comments and Suggestions for Authors
This manuscript reports the results of an interaction study between time-of-day preference and performance on various neuropsychological tests given at preferred or non-preferred times of day in a smallish sample of children. Interestingly, the authors found in addition to the presence of such an interaction that performance was better in the non-preferred time of day for some tasks, and other tasks showed a time of day advantage regardless of preference. These are interesting and in some cases unexpected (to me) findings that merit publication. The paper is nicely written and interesting. The small sample size is my only real concern with the paper, but the authors acknowledge this limitation and as data collection progressed over the pandemic it is an understandable limitation. Nevertheless, a power analysis demonstrating adequate power to detect the sought-after interaction would bolster the reader’s confidence.
Other, mostly quite minor, comments follow.
Introduction:
Line 53: “… albeit being of small magnitude (r = .08)”. If the original paper provides a p-value for that effect, it might be useful to readers to include it here.
Line 61: “ - 5- and 6-year-olds” It looks like you have a stray hyphen on this line.
Methods: In the intro, the authors mention that morningness-eveningness is a continuum. From the initial pool of 483 kids, were any classified as “intermediate”? Was there a deliberate decision to exclude intermediate chronotypes from analysis (which might make sense, given limited power in an interaction study)? If so, it would be helpful for the reason for this exclusion to be mentioned in the methods.
Lines 146-148: Were the children in each group divided into morning and afternoon assessments randomly?
Reliability indices for the neuropsychological tests would be useful.
My biggest (only, really) concern with the paper is the small samples of ET and MT children. I think a power analysis, demonstrating that the authors have sufficient power to detect the interaction of interest, is warranted in the methods.
Results:
Figures 1, 2 and 3: Error bars (SE or 95% CI) would be appropriate on these figures.
Author Response
The authors wish to thank the opportunity to strengthen the manuscript with valuable comments and queries from the Reviewers. We value the careful attention and constructive feedback on our original submission titled “Chronotype, time-of-day, and children's cognitive performance in remote neuropsychological assessment”. Amendments were incorporated to reflect the detailed suggestions the Reviewers have provided. We hope that our edits and the responses provided below adequately address all the issues and concerns the Reviewers have noted.
Reviewer #2
Comment: The small sample size is my only real concern with the paper, but the authors acknowledge this limitation and as data collection progressed over the pandemic it is an understandable limitation.
Response: We wish to thank Reviewer #2 for the valuable feedback. As highlighted, this limitation arose due to the unprecedented challenges encountered during data collection amid the pandemic. We have duly acknowledged this constraint in the manuscript and provided contextual information to elucidate its impact on our findings.
Comment: Nevertheless, a power analysis demonstrating adequate power to detect the sought-after interaction would bolster the reader’s confidence.
Response: We appreciate your insightful suggestion to conduct a post hoc analysis of achieved power to provide readers with a clearer understanding of the study's findings. As detailed in the manuscript, our data collection focused specifically on children with definitely evening or morning chronotypes, resulting in a smaller target population compared to the general population. Additionally, the data collection period coincided with the COVID-19 pandemic, during which remote neuropsychological measures were not feasible for all schools. Consequently, achieving the optimal sample size considering our research goals proved challenging.
Despite these constraints, we were able to involve in the study almost 500 hundred parents/caregivers with 76 of them reporting data that seemed to be in accordance with their children being morning or evening chronotypes. For this sample, considering a moderate effect size (effect size f of .30), the between-subjects design, and the necessity to divide the sample into four groups to address both chronotype and time-of-day effects, and considering the two tails of the distribution due to the lack of clarity in this regard in previous studies, our analysis revealed an achieved (insufficient) power of .732. We have now included this power value in the discussion section to underscore the need for careful consideration of our findings.
Comment: Line 53: “… albeit being of small magnitude (r = .08)”. If the original paper provides a p-value for that effect, it might be useful to readers to include it here.
Response: We would like to thank Reviewer #2 for the careful observation. We have incorporated this information in the revised manuscript to provide a clearer understanding of the statistical significance of the observed effect.
Comment: Line 61: “ - 5- and 6-year-olds” It looks like you have a stray hyphen on this line.
Response: We would like to thank Reviewer #2 for bringing this to our attention. There was, in fact, an unnecessary hyphen in the mentioned line. We have removed it in the revised version of the manuscript.
Comment: In the intro, the authors mention that morningness-eveningness is a continuum. From the initial pool of 483 kids, were any classified as “intermediate”? Was there a deliberate decision to exclude intermediate chronotypes from analysis (which might make sense, given limited power in an interaction study)? If so, it would be helpful for the reason for this exclusion to be mentioned in the methods.
Response: We wish to thank Reviewer #2 for the insightful feedback. We appreciate the suggestion to address the exclusion of intermediate chronotypes in our methods section. As for the rationale for this decision, given the research focus on examining the influence of chronotype and time of day on cognitive performance, we specifically selected children at more extreme ends of the chronotype continuum to better discern potential nuanced variations between individuals from opposite chronotypes. We have included this clarification in the participants' subsection in the revised manuscript.
Comment: Lines 146-148: Were the children in each group divided into morning and afternoon assessments randomly?
Response: We apologize for any confusion, and we appreciate the opportunity to clarify this aspect of our methodology. Participants in each group were indeed divided into morning and afternoon assessments based on random allocation. We have revised the manuscript to mention it in a clearer way in the methods section.
Comment: Reliability indices for the neuropsychological tests would be useful.
Response: We appreciate the suggestion regarding the inclusion of reliability indices for the neuropsychological tests used in our study. We agree that providing reliability indices would enhance the comprehensiveness of our manuscript and have included them in the revised version.
Comment: Figures 1, 2 and 3: Error bars (SE or 95% CI) would be appropriate on these figures.
Response: We appreciate your suggestion to include error bars (SE or 95% CI) on our figures. We agree that this addition would provide readers with a better understanding of the variability in our data and enhance the interpretability of the results. In the revised manuscript, we included error bars on the figures.